# Iron Supplementation Delays Aging and Extends Cellular Lifespan through Potentiation of Mitochondrial Function

**DOI:** 10.3390/cells11050862

**Published:** 2022-03-02

**Authors:** Jovian Lin Jing, Trishia Cheng Yi Ning, Federica Natali, Frank Eisenhaber, Mohammad Alfatah

**Affiliations:** 1Bioinformatics Institute (BII), A*STAR, Singapore 138671, Singapore; linjj@intern.bii.a-star.edu.sg (J.L.J.); chengynt@intern.bii.a-star.edu.sg (T.C.Y.N.); 2Singapore Institute of Food and Biotechnology Innovation (SIFBI), A*STAR, Singapore 138669, Singapore; natali_federica_from.tp@sifbi.a-star.edu.sg; 3School of Biological Sciences (SBS), Nanyang Technological University (NTU), Singapore 637551, Singapore; 4Genome Institute of Singapore (GIS), A*STAR, Singapore 138672, Singapore

**Keywords:** iron, chronological aging, cellular lifespan extension, mitochondria, AMPK, *Saccharomyces cerevisiae*

## Abstract

Aging is the greatest challenge to humankind worldwide. Aging is associated with a progressive loss of physiological integrity due to a decline in cellular metabolism and functions. Such metabolic changes lead to age-related diseases, thereby compromising human health for the remaining life. Thus, there is an urgent need to identify geroprotectors that regulate metabolic functions to target the aging biological processes. Nutrients are the major regulator of metabolic activities to coordinate cell growth and development. Iron is an important nutrient involved in several biological functions, including metabolism. In this study using yeast as an aging model organism, we show that iron supplementation delays aging and increases the cellular lifespan. To determine how iron supplementation increases lifespan, we performed a gene expression analysis of mitochondria, the main cellular hub of iron utilization. Quantitative analysis of gene expression data reveals that iron supplementation upregulates the expression of the mitochondrial tricarboxylic acid (TCA) cycle and electron transport chain (ETC) genes. Furthermore, in agreement with the expression profiles of mitochondrial genes, ATP level is elevated by iron supplementation, which is required for increasing the cellular lifespan. To confirm, we tested the role of iron supplementation in the AMPK knockout mutant. AMPK is a highly conserved controller of mitochondrial metabolism and energy homeostasis. Remarkably, iron supplementation rescued the short lifespan of the AMPK knockout mutant and confirmed its anti-aging role through the enhancement of mitochondrial functions. Thus, our results suggest a potential therapeutic use of iron supplementation to delay aging and prolong healthspan.

## 1. Introduction

The aging population is growing dramatically worldwide [1,2,3]. Aging is the biggest risk factor for several chronic diseases including cancer, neurodegeneration, heart disease, diabetes, cognitive impairment, and immune system decline [4,5,6,7,8]. A gradual decrease in metabolic functions increases the vulnerability to age-related pathologies in the older population that affects health for the remaining life. Recent research in different organisms, from single-celled yeast to animal model systems, has shown that aging mechanisms are conserved and controlled by a highly interconnected and functionally redundant gene and protein interaction network of cellular metabolism [9,10,11].

The budding yeast *Saccharomyces cerevisiae* is a powerful model organism for studying the biology of aging as many of its cellular processes are conserved in humans. Yeast has a tractable short lifespan and is amenable for high-throughput screening under various environmental conditions [9,10,11,12]. Yeast has been extensively used to study human replicative and chronological aging by analyzing its replicative lifespan (RLS) and chronological lifespan (CLS) [13,14,15]. The RLS analyzes the number of times a mother cell divides to form daughter cells, a replicative human aging model for mitotic cells such as stem cells. The CLS is the duration of time that a non-dividing cell is viable in the stationary phase, a chronological human aging model for post-mitotic cells such as neurons.

Ongoing efforts in aging biology research have demonstrated that delaying aging is feasible by modulating the biological processes of aging [11,16]. Currently, the most promising anti-aging interventions are rapamycin and metformin drugs [11,16]. Rapamycin inhibits a highly conserved protein kinase metabolic regulator target of rapamycin complex 1 (TORC1). Metformin increases the adenosine monophosphate-activated protein kinase (AMPK) activity. TORC1 promotes cellular anabolic processes such as the synthesis of proteins, nucleotides, and lipids [17,18]. On the other hand, TORC1 inhibits the catabolic process, including oxidative phosphorylation and autophagy. However, AMPK has opposite metabolic functions as it potentiates catabolic processes and inhibits anabolic processes [19]. Under nutrient-limited conditions, AMPK is activated and inhibits TORC1. The resulting metabolic response increases the cellular energy production by inducing catabolic processes which coordinate with the decrease in ATP utilization in anabolic processes [19,20,21]. Rapamycin and metformin are in clinical trials for their use as anti-aging therapeutics [16,22].

Iron is an important nutrient and essential for almost all organisms, including humans [23,24,25,26]. It plays a crucial role in cellular metabolism and functions as a cofactor for several enzymatic reactions. Since cellular metabolism is the major determinant of aging and iron is crucial for several diverse metabolic functions, we investigated the effect of iron supplementation on aging. We examined the effect of iron supplementation on the CLS of yeast. We found that iron supplementation delays aging and increases the cellular lifespan. We further revealed that the anti-aging mechanism of iron supplementation is through enhancing mitochondrial functions. We found that iron supplementation improves mitochondrial functions and rescues the short lifespan of the mutant of AMPK.

## 2. Materials and Methods

### 2.1. Yeast Strains and Gene Deletion

The prototrophic *Saccharomyces cerevisiae* CEN.PK wild-type strain was used in all experiments to avoid the effects of amino acid auxotrophy on cell growth and survival [27,28]. The gene knockout strain was generated by PCR-mediated homologous recombination whereby the entire locus was replaced by an amplified selection marker containing the upstream and downstream flanking sequences of the target gene [29]. PCR confirmed stable integration of the amplified selection marker.

### 2.2. Medium Composition and Chemicals

The rich medium YPD contained 1% yeast extract, 2% peptone, and 2% glucose, YPD agar (2.5% Bacto agar), and the SD medium contained 6.7 g/L yeast nitrogen base with ammonium sulfate without amino acids (Difco) and 2% glucose. FeSO_4_.7H_2_O, FeCl_3_, CaSO_4_.2H_2_O_,_ MgSO_4_.7H_2_O_,_ CaCl_2_.2H_2_O_,_ MgCl_2_.6H_2_O, and BPS were purchased from Sigma, and their stock solution was prepared in water. Antimycin A (Sigma, St. Louis, MO, USA), DiOC6(3) (3,3′-Dihexyloxacarbocyanine Iodide) (Invitrogen™, Waltham, MA, USA), and MitoTracker™ Deep Red stock (Invitrogen™) solution was prepared in dimethyl sulfoxide (Sigma). The final concentration of DMSO did not exceed 1% in any assay.

### 2.3. Yeast Growth Conditions

The wild-type and deletion strains were recovered from frozen glycerol stock on YPD agar medium at 30 °C. Yeast cells were grown in SD medium overnight at 30 °C with shaking at 220 rpm. The cell cultures grown overnight were diluted to OD600nm~0.2 in fresh SD medium to initiate growth assay in a 96-well plate or flask with or without chemicals and incubated at 30 °C. The cell growth (OD600nm) was measured at different time points using a microplate reader or spectrophotometer.

### 2.4. Chronological Aging Assay

Chronological aging was determined by measuring the chronological lifespan (CLS) of yeast cells, as previously reported with slight modifications [30,31]. The CLS experiment was carried out on a 96-well plate. The overnight cell culture was diluted to OD600nm~0.2 in fresh SD medium and transferred into the 96-well plate containing different concentrations of chemicals and incubated at 30 °C. Cells were grown to a stationary phase which was considered Day1 (100% cell survival) for the CLS analysis. Yeast cells’ survival was quantified at various age time points by an outgrowth assay. Chronologically aged cells (3 μL) at different age time points were transferred to a second 96-well plate. YPD medium (200 μL) was added to a 96-well plate and incubated for 24 h at 30 °C. Cell outgrowth was measured by absorbance (OD600nm) using the microplate reader. Cell survival for different age points was quantified relative to Day1 (considered 100% cell survival) using the calculation formula:Survival % = [Outgrowth OD600nm (age point)/Outgrowth OD600nm (Day1)] × 100

### 2.5. Oxidative Resistance Assay

Yeast cells were grown to stationary phase stage (72 h) in SD medium. After that, cells were washed and diluted to OD600nm~0.2 in a YPD medium with different concentrations of H_2_O_2_ and grown at 30 °C. Oxidative stress resistance was analyzed by comparing the cell growth of H_2_O_2_ treated cells with the non-treated control.

### 2.6. RNA Extraction, cDNA Synthesis, and Quantitative Real-Time PCR

Total RNA was extracted from cells using the RNeasy mini kit (Qiagen, Hilden, Germany) following the manufacturer’s mechanical disruption protocol. RNA concentration and quality were determined by spectrophotometer (NanoDrop 2000 Thermo Scientific, Waltham, MA, USA). Typically, 1 μg total RNA was reverse transcribed into cDNA using QuantiTect Reverse Transcription Kit (Qiagen). During the synthesis of cDNA, two negative controls including without reverse transcriptase and RNA template were also performed. All RT-PCR was performed in a final volume of 20 μL containing 20 ng of cDNA using SYBR Fast Universal qPCR Kit (Kapa Biosystems, Boston, MA, USA) and analyzed using the Quant Studio 6 Flex system (Applied Biosystems, Waltham, MA, USA). The RT-PCR condition was one hold at {95 °C, 180 s}, followed by 40 cycles of {95 °C, 1 s}, and {60 °C, 20 s} steps. After amplification, a melting curve was performed to verify PCR specificity and the absence of primer dimers. The quantitative abundance of each gene was determined relative to the house-keeping transcript ACT1. The relative gene expression between the control and treated conditions was calculated using the 2^−ΔΔCt^ method [32]. A list of primers used for RT-PCR is provided in Appendix A.

### 2.7. Mitochondrial Membrane Potential and Structure Analysis

Yeast cultures were washed with 1× PBS (phosphate buffered saline) and incubated with DiOC6(3) (100 nM) for 30 min in the dark. After incubation, cells were washed and resuspended in 1× PBS. The fluorescence reading (excitation at 482 nm, emission at 504 nm) of samples were measured by the microplate reader. The fluorescence intensity of each sample was normalized with OD600nm. For analysis of mitochondrial structure cells were washed with 1× PBS and incubated with MitoTracker deep red (100 nM) for 30 min in the dark. The samples were visualized with a fluorescence microscope at 100× magnification and the images were processed by ImageJ software.

### 2.8. ATP Analysis

Yeast cultures were mixed with tricholoroacetic acid to a final concentration of 5% and kept on ice for at least 5 min. Next, the cells were spun down, resuspended in 150 μL 5% trichloroacetic acid, and lysed with glass beads on Precellys^®^ 24 homogenizer (Bertin Technologies, Montiny Le Bretonne, France). ATP level was measured using an Enliten luciferin/luciferase reagent kit (Promega, Madison, WI, USA) in a luminometer (Biotek, Winusky, VT, USA). Protein concentration was estimated with Bradford reagent (Bio-rad, Hercules, CA, USA). The ATP level in the samples was normalized to total protein concentration.

### 2.9. Statistical Analysis

All experiments were performed in three independent biological triplicates. Statistical analysis such as mean value, standard deviations, and statistical significance was calculated using GraphPad Prism 9 software.

## 3. Results

### 3.1. Iron Supplementation Extends the Cellular Lifespan of Yeast

We investigated the effect of Iron(II) sulfate (FeSO_4_) supplementation on the chronological lifespan (CLS) of yeast in a 96-well plate. First, cells were incubated with different concentrations of FeSO_4_ in a synthetic defined (SD) medium, and growth was analyzed at different time points (16 h, 24 h, and 48 h). Cell growth reached saturation approximately 24 h after incubating with FeSO_4_ concentrations (Appendix A). Next, we determined the CLS of cells incubated with different concentrations of FeSO_4_. We considered the 48 h stationary phase cell culture as Day1 for CLS analysis. The viability of aged cells at different points was normalized with Day1 (100% viable) and plotted on the survival graph. We found that different concentrations of FeSO_4_ supplementation in the medium extended the CLS of yeast (Figure 1a and Appendix A). We also tested Iron(III) chloride (FeCl_3_) and found a similar result to FeSO_4_ (Figure 1b and Appendix A).

To clarify whether FeSO_4_ and FeCl_3_ salts, their components Iron(II) and Iron(III), or sulfate and chloride extend CLS, we examined the other sulfate and chloride-containing salts. Yeast cells were incubated with CaSO_4,_ MgSO_4,_ CaCl_2,_ MgCl_2_ including FeSO_4,_ and FeCl_3_ in the SD medium. Growth of cells incubated with different salts reached saturation after 24 h (Appendix A). We next measured the survival and found that except FeSO_4_ and FeCl_3_, other sulfate or chloride-containing salts did not extend the lifespan of yeast (Figure 1c). These results revealed that iron, but not sulfate and chloride, extend the CLS of yeast.

For further validation, we depleted the iron and analyzed the CLS of yeast. Bathophenanthrolinedisulfonic acid (BPS) is a specific iron chelator compound that sequesters the iron and leads to iron deficiency in the cells [33]. Yeast cells were incubated with iron and different concentrations of BPS in the SD medium. Cell growth of different concentrations reached saturation after 24 h (Appendix A). We further measured the cell survival and found that BPS addition reduced the lifespan of iron-supplemented cells (Figure 1d). Altogether, these results confirmed that iron supplementation increases the lifespan of yeast.

### 3.2. Iron Supplementation Increases Oxidative Stress Resistance

Cellular aging is associated with increased oxidative stress that damages the biological systems and mitigates age-related diseases [34]. Since iron supplementation delayed aging, we investigated its effect on oxidative stress. We used an oxidative stress inducer compound, hydrogen peroxide (H_2_O_2_), to test the oxidative resistance of cells. Cells were first grown in the presence of iron to stationary phase stage (72 h) in SD medium. After that, cells were washed and incubated with different concentrations of H_2_O_2_ in the YPD medium and grew for 24 h. Oxidative stress resistance was analyzed by comparing the cell growth of the H_2_O_2_ treated cells with the non-treated control. We found that cells supplemented with iron were resistant to oxidative stress compared to control (Figure 2a and Appendix A). To confirm that iron supplementation provides resistance to oxidative stress, BPS was added to iron and cell growth with H_2_O_2_ was analyzed. The addition of BPS reduced the oxidative stress resistance of iron-supplemented cells (Figure 2b). These results correlate with the role of iron supplementation in delaying aging and extending the cellular lifespan.

### 3.3. Iron Supplementation Potentiates Mitochondrial Functions

Mitochondria are the major cellular hub for iron utilization and metabolism [35,36,37,38]. A decline in mitochondrial functions is associated with aging [39,40,41,42,43]. Iron serves as a cofactor of several mitochondrial proteins, including iron-sulfur clusters and heme-containing proteins [44,45,46]. These iron-containing proteins are involved in the mitochondrial tricarboxylic acid (TCA) cycle and electron transport chain (ETC) (Figure 3a). We investigated whether the increase in cellular lifespan by iron supplementation required mitochondrial functions. To test this, we analyzed the expression of mitochondrial TCA cycle genes. We found that iron supplementation significantly induces the expression of several TCA cycle genes (Figure 3b and Appendix A). TCA-cycle metabolites α-ketoglutarate and oxaloacetate can produce glutamate and aspartate by cataplerotic reactions in mitochondria (Figure 3a). These amino acids are utilized in the biosynthesis of proteins, lipids, and nucleotides [47,48,49]. Interestingly, the expression of glutamate (*GDH1* and *GDH3*) and aspartate (*AAT1* and *AAT2*) biosynthetic genes was decreased in iron-supplemented cells (Figure 3c). This expression profile suggests that iron supplementation promotes preservation instead of the consumption of TCA cycle intermediates. In agreement with this idea, gene expression of the mitochondrial anaplerotic pathway genes (*PYC1*, *PYC2*, and *GDH2*) was significantly increased in iron-supplemented cells (Figure 3c). *PYC1* and *PYC2* encode pyruvate carboxylase that converts pyruvate to oxaloacetate. *GDH2* encodes glutamate dehydrogenase which synthesizes α-ketoglutarate from glutamate. Together, these results suggest that iron supplementation configures the cells in a metabolic state that favors anaplerosis and prevents cataplerosis to boost the TCA cycle metabolites.

TCA cycle reactions generate NADH and FADH_2_ which are oxidized by ETC complexes I and II and required for the functionality of the TCA cycle (Figure 3a). Although *S. cerevisiae* lacks complex I, reducing equivalents are transferred to the respiratory chain through NADH dehydrogenases. Succinate dehydrogenase plays a central role and participates in both the TCA cycle and the ETC complex II (Figure 3a). Strikingly, the expression of succinate dehydrogenase (*SDH1* and *SDH2*) was highly upregulated among all analyzed TCA cycle genes (Figure 3b,d). These results indicate that the TCA cycle flux continues towards ETC instead of accumulating a particular intermediate. Likewise, the expression of all other genes of ETC complexes was highly upregulated in iron-supplemented cells (Figure 3d). ETC is associated with the generation of reactive oxygen species (ROS), which regulate the expression of the *SOD2* gene [50]. It encodes a manganese-superoxide dismutase (MnSOD) which is the principal scavenger of mitochondrial superoxide. *SOD2* gene expression was upregulated in iron-supplemented cells (Appendix A) which is consistent with the expression of ETC genes. We further analyzed the mitochondrial membrane potential (MMP) which is generated by the proton pumps of ETC complexes I, III, and IV. We found that iron supplementation increased the MMP of the cells (Appendix A). We then examined the structure of mitochondria using fluorescence microscopy. We found that iron supplementation prevents the fragmentation of mitochondria (Appendix A). Altogether, these results indicate that iron supplementation potentiates the mitochondrial functions of cells.

### 3.4. Iron Supplementation Increases the ATP Level Required for Extension of Cellular Lifespan

Mitochondrial TCA cycle reactions produce reducing equivalents NADH and FADH2 which transfer electrons to ETC and generate adenosine triphosphate (ATP) through oxidative phosphorylation (OXPHOS) (Figure 3a). Since the expression of the TCA cycle and ETC genes were enhanced by iron supplementation, we tested its effect on ATP synthesis. In agreement with the expression profile of the mitochondrial TCA cycle, ETC gene expression and ATP levels were high in iron-supplemented cells (Figure 4a).

We then asked whether ATP is required for the extension of the lifespan of iron-supplemented cells. We inhibited the ATP synthesis and measured the CLS of yeast. Antimycin A (AMA) is an inhibitor of ATP synthesis which binds to complex III and blocks electron transfer in the mitochondrial ETC [51]. We first examined the ATP level and found that AMA treatment inhibits ATP synthesis (Figure 4b). We further tested the effect of AMA on the cellular lifespan of iron-supplemented cells. Yeast cells were incubated with iron and AMA in the SD medium. Growth of cells reached saturation after 24 h (Appendix A). Subsequently, we measured the survival and found that AMA treatment inhibits the iron supplementation mediated extension of lifespan (Figure 4c). Together, these findings reveal that iron supplementation increases the ATP level which is required for the extension of cellular lifespan. We also observed that the lifespan of AMA-treated cells was lower than the control (Figure 4c), further confirming that the inability to synthesize ATP compromised lifespan.

### 3.5. Iron Supplementation Prevents Accelerated Aging of AMPK Knockout Mutant

AMPK is the master regulator of cellular energy homeostasis [20,21]. The highly conserved human analog of AMPK in yeast *S. cerevisiae* is the Snf1 protein [52]. AMPK activates mitochondrial functions to produce ATP under energy-limited conditions. Recent reports have shown that the decline in mitochondrial functions with age occurs in part through the impaired activity of AMPK in different aged organisms [53,54]. Thus, the absence of AMPK activity affects the mitochondrial functions and compromises numerous cellular functions including metabolism, resistance to stress, and cell survival which are the most critical determinants of aging and lifespan. Consistent with previous studies, we found that the *snf1* knockout mutation compromised ATP level, resistance to oxidative stress, and lifespan (Figure 5a–c).

Since the *snf1* mutant is defective in mitochondrial functions, we tested whether iron supplementation can rescue the accelerated aging phenotypes. Therefore, we supplemented the iron to *snf1* mutant and analyzed the ATP level, oxidative stress, and lifespan. We found that iron supplementation increased the ATP level, resistance to oxidative stress, and lifespan (Figure 5d–f). Altogether, these findings confirmed that iron supplementation delays aging and extends cellular lifespan by increasing mitochondria functions.

## 4. Discussion

Nutrients determine the functional status of cells and the deficiency of essential nutrients compromises human health [55,56]. Furthermore, nutrients are the major regulators of cellular metabolism which controls several biological processes including aging, a major risk factor of several chronic diseases. A decline in metabolic activity is one of the hallmarks of aging [10]. Recent research in different organisms, including mammals, has demonstrated that delaying aging and increasing healthspan is feasible by anti-aging interventions including rapamycin and metformin drug administration [11,16]. These drugs target the nutrient-sensing complexes TORC1 and AMPK which are the important metabolic regulators of the cells [11,16].

Iron is an essential nutrient involved in several crucial metabolic reactions in the cells [23,24,25,26]. Iron deficiency impairs metabolic activity resulting in compromised cellular functions, leading to many diseases, including anemia, cognitive impairment, and loss of muscle strength [26,57,58,59]. Iron deficiency is widespread in elderly populations aged ≥65 years [60,61,62].

Since iron regulates metabolic processes, we investigated its role in aging. We utilized yeast as a model organism to examine the role of iron in chronological aging. We investigated the effect of iron supplementation on the CLS of yeast. We found that both FeSO_4_ and FeCl_3_ increased the cellular lifespan. Using different salts of sulfate, chloride, and iron chelator, we confirmed that the cellular lifespan is extended by iron. Aging is related to a gradual accumulation of oxidative stress, which is harmful to cellular functions and decreases cell survival [34]. Since we found that iron supplementation delayed aging, we tested whether it could provide resistance to oxidative stress. To examine the oxidative stress phenotype, we treated the cells with oxidative inducer agent H_2_O_2_ and measured the cell survival. We found that iron supplementation increased oxidative stress resistance compared to control. These findings correlate with the role of iron supplementation in the extension of lifespan.

We further unraveled the anti-aging mechanism of iron supplementation. Mitochondria are the main cellular consumers of iron utilization and metabolism [35,36,37,38]. We first analyzed the expression of mitochondrial TCA cycle genes. We found that the expression of almost all TCA cycle genes was upregulated by iron supplementation. Importantly, iron supplementation downregulated the expression of mitochondrial anaplerosis and upregulated cataplerotic metabolic genes. These results reveal that iron supplementation enhances the synthesis of TCA cycle intermediates and prevents their cellular utilization. These findings supported the anti-aging activity of iron supplementation, as regular export of TCA cycle intermediates affects the mitochondrial integrity [47]. Moreover, replenishing the mitochondrial carbon pool is essential to maintain the mitochondrial functions required for survival during cellular aging.

The TCA cycle intermediate α-ketoglutarate has been shown to extend the lifespan of different organisms [63]. However, we found that iron supplementation increased the expression of α-ketoglutarate dehydrogenase (*KGD1* and *KGD2*) which converts α-ketoglutarate to form succinyl-CoA. Moreover, we observed that the expression of succinate dehydrogenase genes (*SDH1* and *SDH2*) was highly upregulated among other tested genes of the TCA cycle. Importantly, succinate dehydrogenase participates in both the TCA cycle and the ETC complex II. These results suggested that instead of accumulating a particular TCA cycle intermediate, the anti-aging activity of iron supplementation could involve the ETC pathway.

Since the TCA cycle is functionally connected with ETC, we analyzed the expression of ETC genes. We found that iron supplementation highly upregulated the expression of ETC genes. TCA cycle products NADH and FADH_2_ are oxidized by ETC complexes and generate ATP through OXPHOS. We found that iron supplementation increases the cellular ATP level, which is correlated with the upregulation of the TCA cycle and ETC genes. Next, we elucidated whether the increase in ATP level by iron supplementation is required for the extension of cellular lifespan. We found that lifespan extension by iron supplementation was abolished by inhibiting ATP synthesis. Thus, these findings suggest that iron supplementation increases the level of ATP which is required for the extension of cellular lifespan. Our results are consistent with the previous reports showing the role of ATP in the cellular lifespan [51,64,65]. Further, we utilized iron supplementation to enhance the mitochondrial functions and rescued the short lifespan and oxidative stress-sensitive phenotype of the AMPK mutant. Altogether, these results revealed that iron supplementation potentiates the mitochondrial functions that delay aging and increase the lifespan of cells.

Recent studies have shown that iron supplementation restores the mitochondrial defect of lysosome-impaired mutants and prevents mitochondrial decline during aging [66,67]. Thus, our results support the previous findings that iron supplementation improves mitochondrial functions. Interestingly, one of the earlier studies showed that iron supplementation rescued the accelerated replicative aging of lysosome-impaired mutants; however, the effect on the wild-type cells was not included in the report [67]. Nevertheless, our results are correlated with previous findings, and despite different aging models (chronological aging), we found that iron supplementation delays aging and increases cellular lifespan. Thus, collectively, our results and previous reports clearly suggest that iron supplementation could be a potential therapeutic to target aging and increase healthspan.

## Figures and Tables

**Figure 1 cells-11-00862-f001:**
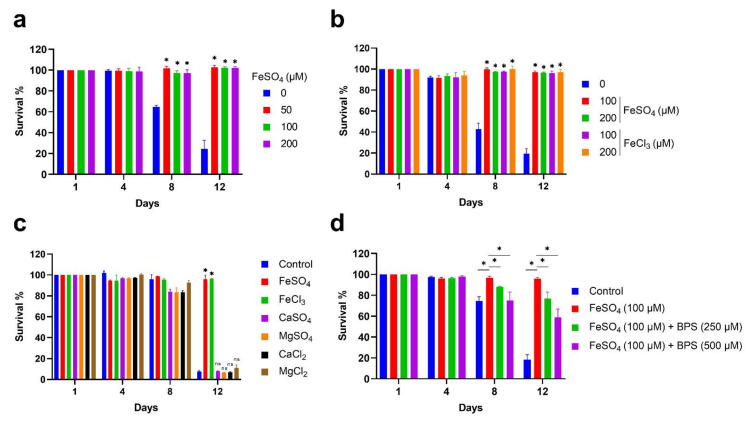
Iron supplementation increases the chronological lifespan of yeast. The prototrophic yeast strain was incubated with different chemical conditions in the synthetic defined (SD) medium and grown in 96-well plates at 30 °C. Cells grown to stationary phase were considered Day1 (100% cell survival) for the chronological lifespan (CLS) analysis. Cell survival was quantified at various age time points by an outgrowth assay. (**a**) CLS of cells supplemented with different concentrations of FeSO4. (**b**) CLS of cells supplemented with different concentrations FeSO_4_ and FeCl_3_. (**c**) CLS of cells supplemented with 100 μM of FeSO_4_, FeCl_3_, CaSO_4_, MgSO_4_, CaCl_2_, and MgCl_2_. (**d**) CLS of cells supplemented with 100 μM of FeSO_4_ in the presence of different concentrations of BPS. Statistical significance (* *p* < 0.05) was determined by Student’s *t*-test.

**Figure 2 cells-11-00862-f002:**
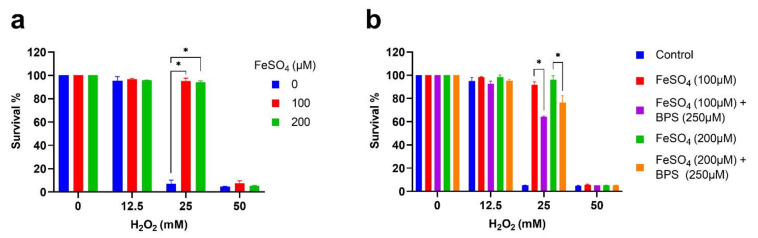
Iron supplementation increases oxidative stress resistance. The prototrophic yeast strain was incubated with different conditions and grown to stationary phase stage in SD medium for 72 h at 30 °C. After that, cells were washed and diluted to OD600nm~0.2 in YPD medium with different concentrations of H_2_O_2_ and grown at 30 °C. (**a**) Oxidative stress analysis for cells incubated with different concentrations of FeSO_4_. (**b**) Oxidative stress analysis for cells incubated with different concentrations of FeSO_4_ and 250 μM BPS. The oxidative stress phenotype was analyzed by comparing the cell growth of H_2_O_2_ treated cells with the non-treated control. Statistical significance (* *p* < 0.05) was determined by Student’s *t*-test.

**Figure 3 cells-11-00862-f003:**
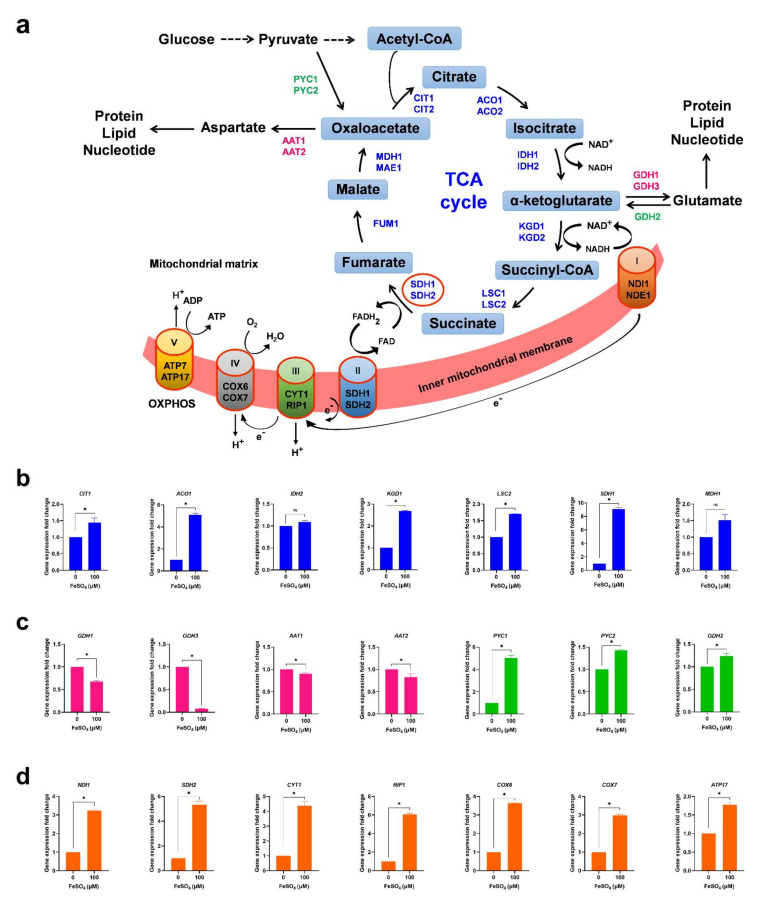
Iron supplementation enhances mitochondrial functions. (**a**) An overview of mitochondrial TCA (tricarboxylic acid) cycle and ETC (electron transport chain) with the major cataplerotic and anaplerotic reactions is illustrated. (**b**) The prototrophic yeast strain was incubated with 100 μM FeSO_4_ and grown in an SD medium for 8 h at 30 °C. RNA was extracted from the cultures and the expression of the indicated TCA cycle genes was analyzed by quantitative RT-PCR. (**c**) Expression analysis of cataplerotic genes (*GDH1*, *GDH3*, *AAT1,* and *AAT2*) and anaplerotic genes (*PYC1*, *PYC2,* and *GDH2*) was done by quantitative RT-PCR. (**d**) Expression analysis of ETC genes was done by quantitative RT-PCR. Statistical significance (* *p* < 0.05) was determined by Student’s *t*-test.

**Figure 4 cells-11-00862-f004:**
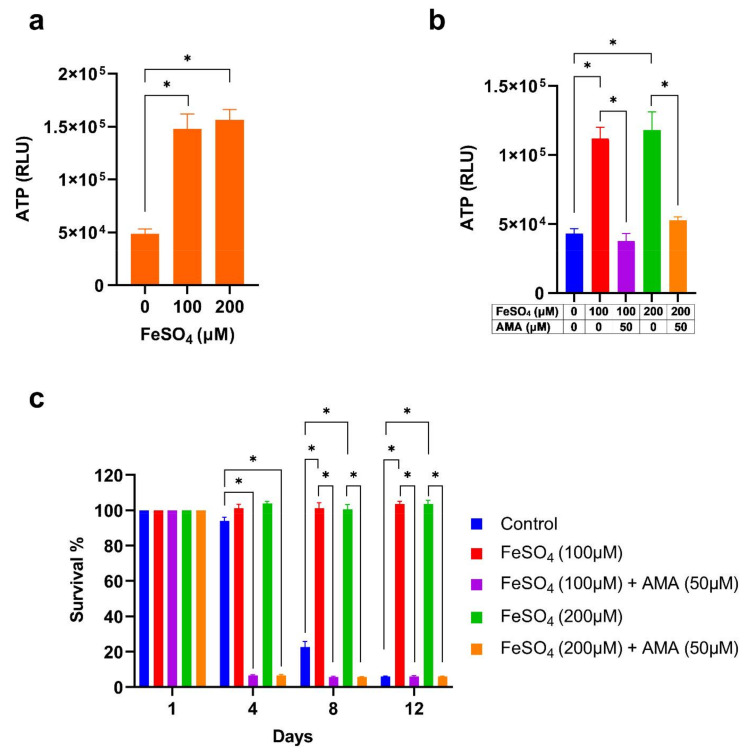
Iron supplementation increases cellular lifespan by enhancing the ATP level. (**a**) The prototrophic yeast strain was incubated with different concentrations of FeSO_4_ and grown to stationary phase stage in SD medium for 72 h at 30 °C. ATP was extracted from the cultures, measured using a luciferin/luciferase reagent in a luminometer, and normalized to total protein concentration. (**b**) ATP analysis of cells incubated with different concentrations of FeSO_4_ and 50 μM antimycin A (AMA). (**c**) Chronological lifespan (CLS) analysis of cells incubated with varying concentrations of FeSO_4_ and 50 μM AMA. Statistical significance (* *p* < 0.05) was determined by Student’s *t*-test.

**Figure 5 cells-11-00862-f005:**
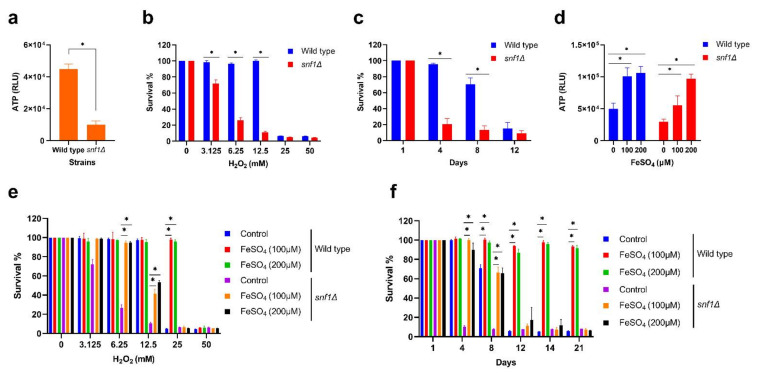
Iron supplementation rescues the accelerated aging of the AMPK knockout mutant. The yeast prototrophic wild-type and AMPK knockout mutant (*snf1∆*) strains were grown to stationary phase in SD medium for 72 h at 30 °C. (**a**) ATP analysis of wild-type and *snf1∆* strains. (**b**) Oxidative stress analysis of wild-type and *snf1∆* strains with different H_2_O_2_ concentrations. (**c**) Chronological lifespan (CLS) analysis of wild-type and *snf1∆* strains. (**d**) ATP analysis of wild-type and *snf1∆* strains incubated with different concentrations of FeSO_4_. (**e**) Oxidative stress analysis of wild-type and *snf1∆* strains incubated with different concentrations of FeSO_4_. (**f**) CLS analysis of wild-type and *snf1∆* strains incubated with different concentrations of FeSO_4_. Statistical significance (* *p* < 0.05) was determined by Student’s *t*-test.

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
