# Peer review of "Iron Supplementation Delays Aging and Extends Cellular Lifespan through Potentiation of Mitochondrial Function"

_cells, 2022, doi:10.3390/cells11050862_

Round 1

Reviewer 1 Report

The current article entitled "Iron supplementation delays aging and extends cellular lifespan through potentiation of mitochondrial function" contains vital information with scientific merits and can be considered for publication.

Overall, the authors showed how iron supplementation enhances the lifespan, and they showed the gene expression analysis of mitochondria, the central cellular hub of iron utilization. Furthermore, quantitative analysis of gene expression data uncovers that iron supplementation upregulates the expression of mitochondrial tricarboxylic acid (TCA) cycle and electron transport chain (ETC) genes.

Next, the authors investigated the role of iron supplementation in the AMPK knockout mutant. They found that iron supplementation rescued the short lifespan of AMPK knockout mutant, which further confirmed the anti-aging role through enhancement of mitochondrial functions and have potential therapeutic use of iron supplementation to delay aging and prolong healthspan. In all, this article is well written.

Few points to be addressed before publication.

There is no p-value in figures 1-5, and the same is with supplementary figures which statistical analysis was performed i.e student t-test, one-way, or two-way ANOVA. Add p-value in all figures.

Do iron supplementation have any effect on mitochondrial membrane potential.

Changes in iron promote ferroptosis and frailty in aging. Does PI-positive dying cells accumulate with aging?.

Does Iron supplementation have any effect on mitochondrial morphological abnormalities? EM or 3 D mitochondrial structure will give some incite.

It is well known that the heme synthesis-export axis modulates oxidative metabolism by regulating the tricarboxylic acid (TCA) cycle flux. The TCA cycle is downmodulated when the axis is enhanced by promoting heme efflux, and oxidative phosphorylation (OXPHOS) decreases. Conversely, when the axis is blocked by either inhibiting heme export or heme synthesis, the TCA cycle and OXPHOS are enhanced by using seashores assay both OCR and ECAR when the iron is supplemented to the aged cells.

Metabolic alterations upon the blockage of heme synthesis-export result in decreased cell proliferation/survival cell cycle result is significant to show them.

A graphical representation or model of these results will be beneficial for readers.

Author Response

Reviewer 2

Comments and Suggestions for Authors

Point 1: The current article entitled "Iron supplementation delays aging and extends cellular lifespan through potentiation of mitochondrial function" contains vital information with scientific merits and can be considered for publication.

Overall, the authors showed how iron supplementation enhances the lifespan, and they showed the gene expression analysis of mitochondria, the central cellular hub of iron utilization. Furthermore, quantitative analysis of gene expression data uncovers that iron supplementation upregulates the expression of mitochondrial tricarboxylic acid (TCA) cycle and electron transport chain (ETC) genes.

Next, the authors investigated the role of iron supplementation in the AMPK knockout mutant. They found that iron supplementation rescued the short lifespan of AMPK knockout mutant, which further confirmed the anti-aging role through enhancement of mitochondrial functions and have potential therapeutic use of iron supplementation to delay aging and prolong healthspan. In all, this article is well written.

Response 1: We thank Reviewer 2 for carefully reading our manuscript.

Few points to be addressed before publication.

Point 2: There is no p-value in figures 1-5, and the same is with supplementary figures in which statistical analysis was performed i.e student t-test, one-way, or two-way ANOVA. Add p-value in all figures. 

Response 2: We thank Reviewer 2 for his/her suggestions. We have included p-value in all figures.

Point 3: Do iron supplementation have any effect on mitochondrial membrane potential. 

Response 3: This is a good point. Mitochondrial membrane potential (MMP) is the energetic indicator of mitochondrial activity. To address this point, we have performed determined the MMP of iron-supplemented cells and compared them with control. We found higher MMP in iron-supplemented cells compared to control. The results are presented in the new supplementary figure 3b. Our conclusions from the experiment remain unchanged.

Point 4: Changes in iron promote ferroptosis and frailty in aging. Does PI-positive dying cells accumulate with aging? 

Response 4: We thank Reviewer 2 for pointing this out. Ferroptosis is a type of programmed cell death that is characterized by the accumulation of lipid peroxidation products and lethal reactive oxygen species (ROS) derived from iron metabolism and can be inhibited by iron chelators. However, we did not observe ferroptosis as iron supplementation increased the viability of chronological aging cells. Moreover, the increase in cells survival was specific to iron supplementation as the addition of iron chelator (BPS) prevented the extension of lifespan (Figure 1d). Although, we are not ruling out the possibilities of ferroptosis phenomenon that could possibly at higher concentrations of iron intake or under different cellular conditions. It would be interesting to identify the mechanism and regulating factors of hormesis effects of iron in the future. We have analyzed the survival of aging cells by outgrowth assay not by PI staining method.

Point 4: Does Iron supplementation have any effect on mitochondrial morphological abnormalities? EM or 3 D mitochondrial structure will give some incite. 

Response 4: We thank Reviewer 2 for his/her points and suggestions. We have not performed the effect of iron supplementation on mitochondrial morphology. However, higher mitochondrial membrane potential (new supplementary figure 3b) indicates that iron supplementation prevents the fragmentation or aggregation of the tubular mitochondrial network. It will be interesting to visualize the mitochondrial structure through EM or 3D, however, to perform the experiments require instruments facilities, and resources which will take time for establishment and validation.

It is well known that the heme synthesis-export axis modulates oxidative metabolism by regulating the tricarboxylic acid (TCA) cycle flux. The TCA cycle is downmodulated when the axis is enhanced by promoting heme efflux, and oxidative phosphorylation (OXPHOS) decreases. Conversely, when the axis is blocked by either inhibiting heme export or heme synthesis, the TCA cycle and OXPHOS are enhanced by using seashores assay both OCR and ECAR when the iron is supplemented to the aged cells. 

Metabolic alterations upon the blockage of heme synthesis-export result in decreased cell proliferation/survival cell cycle result is significant to show them.

Point 4: A graphical representation or model of these results will be beneficial for readers. 

Response 4: We thank Reviewer 2 for his/her suggestions. We have included the model in the graphical abstract.

Reviewer 2 Report

The manuscript entitled "Iron supplementation delays aging and extends cellular lifespan through potentiation of mitochondrial function" by Jing et al describes the observations of the research group on the effects of iron supplementation on the chronological life span (CLS) of Saccharomyces cerevisae strain CEN.PK in Synthetic Defined (SD) medium. They show that supplementation of liquid SD medium with an iron salt (FeSO4 or FeCl3) significantly extends life span and provide associated controls for specificity. CLS extension by iron is associated with increased H2O2/oxidative stress resistance and a transcriptional response of increase of the expression of anaplerotic pathway genes as well as genes encoding electron transport chain components and increased ATP production. The also show that iron supplementation alleviates the negative effects of the SNF1 gene, encoding the catalytic subunit of yeast AMPK, the animal homologue of which has been implicated in aging processes. The authors then discuss their results in context of the potential of iron supplementation to counteract aging.

The work presented is competently done, clearly presented and written in excellent English. However, the implications of the results for our understanding of aging and a role for iron supplementation in prevention of aging processes is vastly overstated. In a nutshell, what the authors have shown is that standard SD media formulations have suboptimal iron concentrations for yeast growth over prolongued incubation times. Compared to rich YPD media and compared to the concentrations of other salts in the YNB formulations used to make SD media, the iron content in the media used for growth of the yeast strain is very low (200 ug/L, about 1,3 uM) (see also reference 66. in the manuscript/Hughes et al Cell 2020). Standard iron content in SD media therefore appears to be insufficient for yeast cells to maintain a functional respiratory chain, which explains many of the results presented here. The most interesting observations (see below) are not explored.

In addition, there are literally hundreds if not thousands of publications on the importance of a balanced iron content to human health and to the elderly, so from the aging point of view this paper has no real novelty.

There are some interesting aspects to the work, for example the increased resistance to oxidative stress, as well as the beneficial aspects of iron supplementation to the snf1 mutant strain. But all in all I am of the opinion that this manuscript in the current form should be published in a more specialized journal focusing on yeast growth rather than in Cells journal, and I would recommend not to accept on grounds that this is just not interesting enough for the broad reader base of Cells.

More specific comments on the data:

  1. I am missing an examination on the threshold level of iron concentration that exerts the beneficial effecs of iron on growth. Figure 1 shows that already 50 uM iron extends CLS as well as 100 or 200 uM, so it would be interesting to find out what concentration is sufficient. FeSO4 can react with water to form H2O2, so higher iron levels may actually not be optimal. This leads to comment:
  2. Do the authors have considered that, in addition to supporting sustained activity of the respiratory chain, higher iron content may prime the cells to react more efficiently towards oxidizing stress (Fig. 2) because of increased H2O2 in the media? The authors show that SOD2 is upexpressed in presence of higher iron levels. This may be just a reflection of a more active respiratory chain, but could be explored e.g. by using smaller amount of Fe supplement and observing the reaction of the culture
  3. Figure 3 doesn't technically show that mitochondrial functions are enhanced upon iron supplementation. It only shows that levels of transcripts encoding proteins with a role in mitochondrial functions are up. So the figure title is overstating the result
  4. Figure 4a and b can be reduced to one panel, as the first three bars of panel b basically show the same result as panel a. Is Figure 4 c missig a dataset? I can't see the graph for FeSO4 (200 uM) + AMA.
  5. The response of the snf1 KO mutation (Fig. 5) is interesting and could be explored a bit more. What is the mechanism of growth rescue? It can't be just ATP, because figure 5 d shows that ATP is near wild type level, althogh  growth is not fully rescued, and the cells do not to better in terms of CLS compared to the 100 uM culture, although the ATP content for the latter is about half. And why is there a clear difference in ATP production between the 100uM and 200 uM cultures? I think these are the most intriguing results and could be expanded on both experimentally as well as in the discussion, where AMPK function seems barely an afterthought.
  6. I am not familiar with the referencing style. Does this fulfill Cells journal standards? There is important info on issue, volume, page number missing, and DOIs seem to be haphazardly thrown in in some places, but not in others

Author Response

Reviewer 3

Comments and Suggestions for Authors

Point 1: The manuscript entitled "Iron supplementation delays aging and extends cellular lifespan through potentiation of mitochondrial function" by Jing et al describes the observations of the research group on the effects of iron supplementation on the chronological life span (CLS) of Saccharomyces cerevisae strain CEN.PK in Synthetic Defined (SD) medium. They show that supplementation of liquid SD medium with an iron salt (FeSO4 or FeCl3) significantly extends life span and provide associated controls for specificity. CLS extension by iron is associated with increased H2O2/oxidative stress resistance and a transcriptional response of increase of the expression of anaplerotic pathway genes as well as genes encoding electron transport chain components and increased ATP production. The also show that iron supplementation alleviates the negative effects of the SNF1 gene, encoding the catalytic subunit of yeast AMPK, the animal homologue of which has been implicated in aging processes. The authors then discuss their results in context of the potential of iron supplementation to counteract aging.

Response 1: We the Reviewer 2 for carefully reading our manuscript.

Point 2: The work presented is competently done, clearly presented and written in excellent English. However, the implications of the results for our understanding of aging and a role for iron supplementation in prevention of aging processes is vastly overstated. In a nutshell, what the authors have shown is that standard SD media formulations have suboptimal iron concentrations for yeast growth over prolongued incubation times. Compared to rich YPD media and compared to the concentrations of other salts in the YNB formulations used to make SD media, the iron content in the media used for growth of the yeast strain is very low (200 ug/L, about 1,3 uM) (see also reference 66. in the manuscript/Hughes et al Cell 2020). Standard iron content in SD media therefore appears to be insufficient for yeast cells to maintain a functional respiratory chain, which explains many of the results presented here. The most interesting observations (see below) are not explored.

Response 2: We thank Reviewer 2 for his/her comments. As reviewer also noted that we worked with prototrophic yeast CEN.PK strain to performed chronological aging and identify anti-aging mechanism of iron supplementation. We utilized CEN.PK strain to study the aging in yeast to avoid the effects of amino acid auxotrophy on cell growth and survival during chronological aging.  Moreover, we used a standard SD medium (YNB with ammonium sulfate without amino acids) which is sufficient to grow the cells. Use of synthetic complete (SC) medium (YNB with ammonium sulfate with amino acids) or undefined YPD rich medium may affect and bring the complexity for proper investigation of the role of iron supplementation in chronological aging due to the contribution of amino acids and undefined constituents from SC and YPD medium respectively.

Point 2: In addition, there are literally hundreds if not thousands of publications on the importance of a balanced iron content to human health and to the elderly, so from the aging point of view this paper has no real novelty.

Response 2: I totally agree with Reviewer 2 that thousands of publications on the importance of balanced iron content to human health and to the elderly. However, I am totally disagreeing with Reviewer 2 that our findings are not novel. Recently, two papers show the importance of iron in maintaining mitochondrial functions (Hughes et al Cell 2020 and Chen KL et al GeroScience 2020). We already cited these papers and even Reviewer 2 mentioned in his/her comments about Hughes et al Cell 2020. Chen KL et GeroScience 2020 has shown that iron supplementation prevents the accelerated replicative aging vacuole/lysosome-impaired mutants. We have provided a detailed discussion about these two papers and compared them with our findings. Together, our manuscript first time investigated the role of iron supplementation in extending chronological lifespan and identified the anti-aging mechanisms of iron.

Point 3: There are some interesting aspects to the work, for example the increased resistance to oxidative stress, as well as the beneficial aspects of iron supplementation to the snf1 mutant strain. But all in all I am of the opinion that this manuscript in the current form should be published in a more specialized journal focusing on yeast growth rather than in Cells journal, and I would recommend not to accept on grounds that this is just not interesting enough for the broad reader base of Cells.

Response 3: We thank Reviewer 2 for his/her comments. Reviewer 2 also agrees that our manuscript has several interesting findings including iron supplementation increased chronological lifespan, the mechanism is through mitochondria, providing resistance to oxidative stress and rescuing the accelerated aging of snf1 mutant. But we were surprised to see Reviewer 2 decision that our manuscript is suitable for a more specialized journal focusing on yeast growth. However, we didn’t observe the beneficial role of iron supplementation in yeast growth (supplementary figures 1 and 2). Our work is completely focused on aging biology and we investigated the role of iron supplementation in aging in yeast, which is a powerful model organism for understanding the cellular aging of humans.

More specific comments on the data:

Point 4: I am missing an examination on the threshold level of iron concentration that exerts the beneficial effects of iron on growth. Figure 1 shows that already 50 uM iron extends CLS as well as 100 or 200 uM, so it would be interesting to find out what concentration is sufficient. FeSO4 can react with water to form H2O2, so higher iron levels may actually not be optimal. This leads to comment:

Response 4: I responded in the previous comments that we didn’t observe the beneficial effects of iron on cells growth (supplementary figures 1 and 2). However, it is interesting to point by Reviewer 2 that what is the lowest concentration of iron is sufficient to extend lifespan. We had tested the lowest iron concentration (6.25uM) which was sufficient to extend the lifespan. The results are presented in the new supplementary figure 1b for effect on cell growth and new supplementary figure 1c for CLS.

Since we tested a range of iron concentrations including 400uM which is also extending the lifespan new supplementary figure 1c. So, this range is optimal as iron has a positive effect on cells survival. To my knowledge, there is no such report that FeSO4 reacts with water and form H2O2. However, Ferrous iron (II) is oxidized by hydrogen peroxide to form ferric iron (III), hydroxyl radical, and water. This reaction is well known as Fenton reaction chemistry.

Point 5: Do the authors have considered that, in addition to supporting sustained activity of the respiratory chain, higher iron content may prime the cells to react more efficiently towards oxidizing stress (Fig. 2) because of increased H2O2 in the media? The authors show that SOD2 is upexpressed in presence of higher iron levels. This may be just a reflection of a more active respiratory chain, but could be explored e.g. by using smaller amount of Fe supplement and observing the reaction of the culture.

Response 5: We thank Reviewer 2 for his/her comments. As responded in the previous comment that we had tested a range of iron concentrations including 400uM. Even this higher tested concentration (400uM) was also efficiently extended the lifespan (new supplementary figure 1c). Importantly and interestingly iron supplemented cells were resistant to oxidative inducer H2O2 toxicity. So, the results clearly indicate that tested iron concentrations have beneficial effects on delaying aging and extending chronological lifespan. However, as for suggestion by Reviewer 2, we tested the H2O2 toxicity of cells supplemented with lower concentrations of iron and found consistent results correlating with the CLS extending concentrations of iron. The results are presented in the new supplementary figure 2. Our conclusions from the experiment remain unchanged.

Point 6: Figure 3 doesn't technically show that mitochondrial functions are enhanced upon iron supplementation. It only shows that levels of transcripts encoding proteins with a role in mitochondrial functions are up. So the figure title is overstating the result

Response 6: We thank Reviewer 2 for his/her comments. We have changed the figure to  “Iron supplementation enhances mitochondrial gene expression”.

Point 7: Figure 4a and b can be reduced to one panel, as the first three bars of panel b basically show the same result as panel a. Is Figure 4 c missig a dataset? I can't see the graph for FeSO4 (200 uM) + AMA.

Response 7:  We thank Reviewer 2 for his/her suggestions. However, figure 4a and 4b belong to different questions and are independently performed. Moreover, new figure 4b changed the sequence to the included p-value.  We have changed the graph style of figure 4c and now FeSO4 (200 uM) + AMA clearly recognizable.

Point 8: The response of the snf1 KO mutation (Fig. 5) is interesting and could be explored a bit more. What is the mechanism of growth rescue? It can't be just ATP, because figure 5 d shows that ATP is near wild type level, althogh  growth is not fully rescued, and the cells do not to better in terms of CLS compared to the 100 uM culture, although the ATP content for the latter is about half. And why is there a clear difference in ATP production between the 100uM and 200 uM cultures? I think these are the most intriguing results and could be expanded on both experimentally as well as in the discussion, where AMPK function seems barely an afterthought.

Response 8:  We thank Reviewer 2 for his/her suggestions and for pointing out the interesting finding associated with iron and snf1 mutant. However, we would like to give a note to Reviewer 2 that we worked on chronological lifespan not on growth and we never mentioned in our manuscript that iron rescued the growth of snf1 KO mutation. But if Reviewer 2 is concerned about chronological lifespan then we can clearly see that iron supplementation rescued the accelerated aging of snf1 mutant to the wild-type control. However, in this concern, I totally agree with Reviewer 2 comment that iron phenotype is not similar to wild type even though ATP level is nearly the same. However, I would like to point out the oxidative stress sensitivity phenotype of snf1 mutant. Iron didn’t fully rescue the oxidative stress sensitivity of snf1 mutant which could be the possibility of only partial recovery of accelerated aging of snf1 mutant by iron supplementation. However, we cannot rule out the possibilities of other snf1 mutant phenotypes such as defective autophagy and nutrient signaling that could be associated with its short lifespan and not suppressed by iron supplementation. It will be interesting to identify the whole cellular response of iron supplementation in aging experimental conditions that map the biological interactions of different pathways of iron in the cells. However, this study would be independently reported if started and completed in the future.

Point 9: I am not familiar with the referencing style. Does this fulfill Cells journal standards? There is important info on issue, volume, page number missing, and DOIs seem to be haphazardly thrown in in some places, but not in others.

Response 9:   We thank Reviewer 2 for pointing out. This would be finally checked by the Cells journal publication team.

Reviewer 3 Report

In the presented study, the authors analyzed the effect of iron on chronological life span of the model organism Saccharomyces cerevisiae. The manuscript is interesting as it shows that iron supplementation delays aging and increases life span through enhancement of mitochondrial function. The authors experimentally proved that addition of iron upregulates the expression of mitochondrial tricarboxylic acid (TCA) cycle and electron transport chain (ETC) genes which in turn led to elevation of ATP level resulting in the increased cellular lifespan.

Major comments:

Although results of the study clearly indicate the positive effect of iron supplementation on life span of the yeast cells, I highly recommend using statistical evaluation to confirm significance of the observed differences.

Moreover, some graphs are unclear, not all data are obvious or clearly recognizable, as for example the Figure 1a, b, c, or the Figure 4c. For sake clarity, I recommend using different style of the graph to represent obtained data.

In the materials and methods, paragraph 2.4. Chronological aging assay, the calculation formula of cell survival would be helpful for better understanding of the used methodology.

Authors indicate that iron supplementation reduces oxidative stress of aging cells. As excessive formation of ROS is the prominent indicator of oxidative stress, it would be interesting to see if iron addition also reduces ROS overproduction. If possible, could the authors add data revealing production of ROS of aging cells before and after Fe addition in the presence or absence of H2O2?

line 139 Statistical analyses. All experiments were performed at least in triplicate on different days. Could authors explain this in more details? How many independent experiments were performed and how many biological or chemical replicates were used or performed?

Minor comments:

line 95 instead of chemicals use sulfate and chloride-containing salts

line 103 Cells were grown to stationary phase “WHICH” was considered…

line 230 figure legend to Figure 3c, in the graph the expression of GDH2 is indicated while in the figure capture it stays GDH1

line 284 …compromised ATP level, “RESISTANCE TO” oxidative stress, and…

line 297 …ATP level, oxidative stress, “AND” lifespan.

line 298 …increased the ATP level, oxidative stress “RESISTANCE”, and lifespan…

supplementary Fig. 2 In the figure legend to the Fig.2 is written: …ETC genes NDE1 and ATP17…, while in the graph the expression of ATP7 is shown

Author Response

Reviewer 4

Comments and Suggestions for Authors

Point 1: In the presented study, the authors analyzed the effect of iron on chronological life span of the model organism Saccharomyces cerevisiae. The manuscript is interesting as it shows that iron supplementation delays aging and increases life span through enhancement of mitochondrial function. The authors experimentally proved that addition of iron upregulates the expression of mitochondrial tricarboxylic acid (TCA) cycle and electron transport chain (ETC) genes which in turn led to elevation of ATP level resulting in the increased cellular lifespan.

Response 1: We thank Reviewer 4 for carefully reading our manuscript.

Major comments:

Point 2: Although results of the study clearly indicate the positive effect of iron supplementation on life span of the yeast cells, I highly recommend using statistical evaluation to confirm significance of the observed differences.

Response 2: We thank Reviewer 4 for his/her suggestions. We have included p-value in all figures.

Point 3: Moreover, some graphs are unclear, not all data are obvious or clearly recognizable, as for example the Figure 1a, b, c, or the Figure 4c. For sake clarity, I recommend using different style of the graph to represent obtained data.

Response 3: We thank Reviewer 4 for his/her suggestions. We have changed the graph style of Figures 1a, b, c, and Figure 4c. Moreover, to keep a similar graph style we change figures 1d, 5c, and 5f.

Point 4: In the materials and methods, paragraph 2.4. Chronological aging assay, the calculation formula of cell survival would be helpful for better understanding of the used methodology.

Response 4: We thank Reviewer 4 for his/her suggestions. We have included the calculation formula in the chronological aging assay method section.

Point 5: Authors indicate that iron supplementation reduces oxidative stress of aging cells. As excessive formation of ROS is the prominent indicator of oxidative stress, it would be interesting to see if iron addition also reduces ROS overproduction. If possible, could the authors add data revealing production of ROS of aging cells before and after Fe addition in the presence or absence of H2O2?

Response 5: We thank Reviewer 4 for his/her comments. We already discussed in the manuscript that iron supplementation increased the expression of TCA and ETC genes. ETC is the major source for ROS that generates reactive oxygen radicals which is converted to another ROS H2O2 by SOD2. So, one could expect that iron-supplemented cells would have more ROS compared to the control cells. We already discussed in the manuscript (previous line 241 -245) and now (revised line 254-258) that based on this cellular reflection SOD2 gene expression was upregulated in iron-supplemented cells. Moreover, iron is involved in the Fenton reaction and Ferrous iron (II) is oxidized by H2O2 to form ferric iron (III), hydroxyl radical, and water. This reaction is well known as Fenton reaction chemistry. Hydroxyl radical is also another ROS that could lead to the toxic lethal effect on cells. Although iron is known to be involved in the generation of ROS, we did not observe the toxic effect of tested iron concentrations on cells survival. Interestingly, iron-supplemented cells increased the viability of chronological aging cells. However, we do not rule the possibilities of oxidative stress adaptation mediated extension of lifespan in iron supplementation cells.

Point 6: line 139 Statistical analyses. All experiments were performed at least in triplicate on different days. Could authors explain this in more details? How many independent experiments were performed and how many biological or chemical replicates were used or performed?

Response 6: We thank Reviewer 4 for his/her comments. All experiments were performed in three independent biological triplicates.

Minor comments:

Point 7: line 95 instead of chemicals use sulfate and chloride-containing salts

Response 7: We thank Reviewer 4 for his/her suggestions. We performed experiments with different chemicals other than sulfate and chloride-containing salts such Antimycin, BPS, etc. So, we did not change and retain the chemicals in the manuscript.

Point 8: line 103 Cells were grown to stationary phase “WHICH” was considered…

Response 8: Done

Point 8: line 230 figure legend to Figure 3c, in the graph the expression of GDH2 is indicated while in the figure capture it stays GDH1

Response 8: revised to GDH2

Point 9: line 284 …compromised ATP level, “RESISTANCE TO” oxidative stress, and…

Response 9: Done

Point 10: line 297 …ATP level, oxidative stress, “AND” lifespan.

Response 10: Done

Point 11: line 298 …increased the ATP level, oxidative stress “RESISTANCE”, and lifespan…

Response 11: Done

Point 12: supplementary Fig. 2 In the figure legend to the Fig.2 is written: …ETC genes NDE1 and ATP17…, while in the graph the expression of ATP7 is shown

Response 12: revised to ATP7

Round 2

Reviewer 1 Report

The authors have addressed previous concerns, but the authors failed to show any data or explanation of the bellow comment.
It is well known that the heme synthesis-export axis modulates oxidative metabolism by regulating the tricarboxylic acid (TCA) cycle flux. The TCA cycle is downmodulated when the axis is enhanced by promoting heme efflux, and oxidative phosphorylation (OXPHOS) decreases. Conversely, when the axis is blocked by either inhibiting heme export or heme synthesis, the TCA cycle and OXPHOS are enhanced by using seashores assay both OCR and ECAR when the iron is supplemented to the aged cells.
Again, it is crucial to show mitochondrial morphological abnormalities. EM or 3 D mitochondrial structure will give some incite. Immunofluorescent confocal images of TOM20 after iron supplementation will give some incite. 

Author Response

Reviewer: Comments and Suggestions for Authors

Point 1: The authors have addressed previous concerns, but the authors failed to show any data or explanation of the bellow comment.

It is well known that the heme synthesis-export axis modulates oxidative metabolism by regulating the tricarboxylic acid (TCA) cycle flux. The TCA cycle is downmodulated when the axis is enhanced by promoting heme efflux, and oxidative phosphorylation (OXPHOS) decreases. Conversely, when the axis is blocked by either inhibiting heme export or heme synthesis, the TCA cycle and OXPHOS are enhanced by using seashores assay both OCR and ECAR when the iron is supplemented to the aged cells.

Again, it is crucial to show mitochondrial morphological abnormalities. EM or 3 D mitochondrial structure will give some incite. Immunofluorescent confocal images of TOM20 after iron supplementation will give some incite.

Response1: In the 1st revised manuscript comment, the Reviewer didn’t ask for any data or explanation and we didn't reply as we thought Reviewer shared the information. Although, Reviewer didn't mention about paper, however, we are already familiar with the paper as the Reviewer pointed us "The heme synthesis-export system regulates the tricarboxylic acid cycle flux and oxidative phosphorylation, Veronica Fiorito et al, Cell report 2021"

[https://www.sciencedirect.com/science/article/pii/S2211124721006173].

We did not work on heme synthesis and regulation and its effect on chronological aging or lifespan.

However, I guess, the Reviewer wanted to perform us seahorse assay to identify the oxygen consumption rate (OCR) and extracellular acidification rate (ECAR) in iron-supplemented cells.

The Seahorse is a tool/machine used to analyze the OCR and ECAR which is the indirect approach to determine the mitochondrial functions such as ATP synthesis. I would like to note here for Reviewer that we have directly quantified several mitochondrial functions such as ATP synthesis, mitochondrial potential in iron supplemented aged cells. We also showed that inhibition of ATP synthesis by using Antimycin A prevents the increased lifespan in iron-supplemented cells.

Moreover, as per the Reviewer's suggestion, we included the result of mitochondrial structure in iron-supplemented cells (Supplementary figure 4c). We examined the mitochondria structure using a well-known mitochondrial-specific fluorescent dye Mitotracker Deep Red. Analysis of mitochondrial structure using EM, 3 D or tagging mitochondrial proteins such as TOM20 is not possible in given 10 days revision time. However, the conclusion of the result will be the same for all the experiments.

Reviewer 2 Report

I appreciate the improvements the authors have made to the manuscript. As stated before, the work is competently done and the data clearly presented. The authors' commentary on my remark Fe(II) and H2O generating H2O2 (vs Fe(II) and Hso2 generating is correct; I should have reviewed this instead of haphazardly stating this from bad memory. Nevertheless, my statement that FeSO4 supplementation has been associated with induction of oxidative stress is correct.

My main concern with this manuscript is the concept of "chronological life span" in yeast, which to me is a semantic slight of hand. What is really measured is growth recovery of yeast cells after prolonged starvation and crowding stress, and in my opinion has little in common with aging processes in humans. I still feel the result is mostly relevant for researchers that work with yeast. However, I am aware that there are many publications that make use of the CLS concept to tie their work into an "aging" narrative.

As the work per se is well done (as stated above) and the results are interesting, I have changed my recommendation to a reluctant accept. I leave it to the editor to make a final decision.

Author Response

Reviewer: Comments and Suggestions for Authors

Point 1: I appreciate the improvements the authors have made to the manuscript. As stated before, the work is competently done and the data clearly presented. The authors' commentary on my remark Fe(II) and H2O generating H2O2 (vs Fe(II) and Hso2 generating is correct; I should have reviewed this instead of haphazardly stating this from bad memory. Nevertheless, my statement that FeSO4 supplementation has been associated with induction of oxidative stress is correct.

My main concern with this manuscript is the concept of "chronological life span" in yeast, which to me is a semantic slight of hand. What is really measured is growth recovery of yeast cells after prolonged starvation and crowding stress, and in my opinion has little in common with aging processes in humans. I still feel the result is mostly relevant for researchers that work with yeast. However, I am aware that there are many publications that make use of the CLS concept to tie their work into an "aging" narrative.

As the work per se is well done (as stated above) and the results are interesting, I have changed my recommendation to a reluctant accept. I leave it to the editor to make a final decision.

Response 1: We thank the Reviewer for reading our revised manuscript. Reviewer is correctly pointed out, the relationship of the "chronological lifespan" concept in yeast to cellular aging is a complex matter. What the experiments really measure is cell survival and, possibly, growth recovery of aged yeast cells after prolonged starvation and crowding stress.

Aging in yeast is assayed primarily by measuring replicative or chronological lifespan. A replicative lifespan (RLS), defined as the number of daughters produced by each dividing mother cell, and a chronological life span (CLS), defined as the capacity of stationary (Go) cultures to maintain viability over time. The CLS assay has been proposed to reflect aging in post-mitotic mammalian cells, such as neurons and muscles cells.

Certainly, not all aspects of post-mitotic cells of yeast culture in the stationary phase (nutrient starvation cause cells enter to the post-mitotic stage) are well reflected with mammalian post-mitotic cells aging here and it makes sense to think about alternative experimental design. This will become an important area for future research to study cellular aging in the yeast model systems.

Despite the different levels of complexity between yeast and human post-mitotic cells, yeast CLS models of aging have been instrumental for the identification of essential conserved pathways that influence healthspan and lifespan. Two of the major pathways studied in the context of aging and age-related disease are the Sirtuin pathway and the TOR signaling pathway, and yeast was pivotal in their discovery. Saccharomyces cerevisiae has directly or indirectly contributed to the identification of arguably more mammalian genes that affect aging than any other model organism. For example, studies aiming to understand the molecular mechanisms of calorie-restriction (CR)-mediated longevity, allowed for the identification of several longevity genes. In yeast, CR down-regulates the conserved Ras/cAMP/PKA, TOR, and Sch9 signaling pathways that integrate the nutrient and other environmental cues to regulate cell growth, division, and lifespan.

We have already provided all the references in the manuscript. The reviewer can also look at the recent publications on yeast CLS in MDPI Cells Journal; Section: Cellular Aging; Special Issue: Yeast as a Model in Aging Research.

  • Flavonoids from Sacred Lotus Stamen Extract Slows Chronological Aging in Yeast Model by Reducing Oxidative Stress and Maintaining Cellular Metabolism; Cells 2022
  • Identification of Tropical Plant Extracts That Extend Yeast Chronological Life Span; Cells 2021

Reviewer 3 Report

The manuscript has improved following revision. In my opinion it is suitable for acceptance in its present form.

Author Response

Reviewer: Comments and Suggestions for Authors

Point1: The manuscript has improved following revision. In my opinion it is suitable for acceptance in its present form.

Response 1: We thank the Reviewer for reading our revised manuscript and recommending acceptance in the Cells journal.